# Variational Classification:
# A Probabilistic Generalization of the Softmax Classifier

**Shehzaad Dhuliawala**                                   *shehzaad.dhuliawala@inf.ethz.ch*
*Department of Computer Science, ETH Zurich, Switzerland*

**Mrinmaya Sachan**                                       *mrinmaya.sachan@inf.ethz.ch*
*Department of Computer Science, ETH Zurich, Switzerland*

**Carl Allen**                                            *carl.allen@ai.ethz.ch*
*AI Centre, ETH Zurich, Switzerland*

## Abstract

We present a latent variable model for classification that provides a novel probabilistic interpretation of neural network softmax classifiers. We derive a variational training objective, analogous to the evidence lower bound (ELBO) used to train variational auto-encoders, that generalises the cross-entropy loss used to train classification models. Treating inputs to the softmax layer as samples of a latent variable, our abstracted perspective reveals a potential inconsistency between their *anticipated* distribution, required for accurate label predictions to be output, and their *empirical* distribution found in practice. We augment the variational objective to mitigate such inconsistency and encourage a chosen latent distribution, instead of the implicit assumption found in a standard softmax layer. Overall, we provide new theoretical insight into the inner workings of widely-used softmax classifiers. Empirical evaluation on image and text classification datasets demonstrates that our proposed approach, *variational classification*[1], maintains classification accuracy while the reshaped latent space improves other desirable properties of a classifier, such as calibration, adversarial robustness, robustness to distribution shift and sample efficiency useful in low data settings.

## 1 Introduction

Classification is a central task in machine learning, used to categorise objects (Klasson et al., 2019), provide medical diagnoses (Adem et al., 2019; Mirbabaie et al., 2021), or identify potentially life-supporting planets (Tiensuu et al., 2019). Classification also arises in other learning regimes, e.g. to select actions in reinforcement learning, distinguish positive and negative samples in contrastive learning, and pertains to the *attention* mechanism in transformer models (Vaswani et al., 2017). Classification is commonly tackled by training a neural network with a *sigmoid* or *softmax* output layer.[2] Each data sample $x$ is mapped deterministically by an *encoder* $f_\omega$ (with weights $\omega$) to a real vector $z = f_\omega(x)$, which the softmax layer maps to a distribution over class labels $y \in \mathcal{Y}$:

$$p_\theta(\mathrm{y}|x) = \frac{\exp\{z^\top w_y + b_y\}}{\sum_{y' \in \mathcal{Y}} \exp\{z^\top w_{y'} + b_{y'}\}} \ . \tag{1}$$

Softmax classifiers have achieved impressive performance (e.g. Krizhevsky et al., 2012), however they are known to suffer from several issues. For example: such classifiers are trained to numerically minimise a loss function over a random dataset and their resulting predictions are *hard to explain*; model predictions may accurately identify the correct class by their mode but less accurately reflect a meaningful class distribution $p(y|x)$, known as *miscalibration*; predictions can vary materially and erroneously for imperceptible changes in the data (*adversarial examples*); and highly flexible neural networks are often used in order to achieve accurate predictions, which tend to *require considerable labelled data* to train.

---

[1] Code: `www.github.com/shehzaadzd/variational-classification`. Review: `www.openreview.net/forum?id=EWv9XGOpB3`
[2] We refer throughout to the softmax function since it generalises sigmoid to multiple classes, but arguments apply to both.

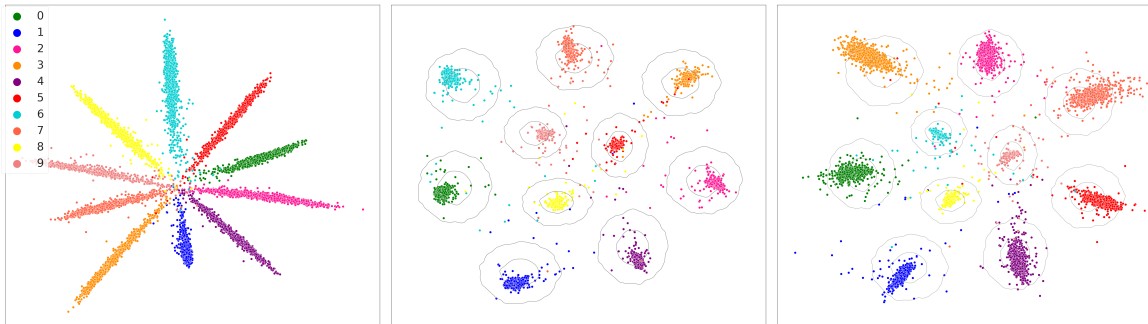

Figure 1: Empirical distributions of inputs to the output layer $q_\phi(z|y)$ for classifiers trained under incremental components of the VC objective (Eqn. 7) on MNIST (*cf* the central $\mathcal{Z}$-plane in figure 2). (*l*) "MLE" objective = softmax cross-entropy; (*c*) "MAP" objective = MLE + Gaussian class priors $p_\theta(z|y)$ (in contour); (*r*) VC objective = MAP + entropy of $p_\theta(z|y)$. Colour indicates class $y$; $\mathcal{Z} = \mathbb{R}^2$ for visualisation purposes.

In order to better understand softmax classification and ideally mitigate some of its known shortcomings, we take a latent perspective, introducing a latent variable z in a graphical (Markov) model y → z → x. This model can be interpreted generatively as first choosing a sample's class, or *what* it is ($y$); then parameters defining its attributes, e.g. size, colour ($z$); which determine the observation ($x$), subject to any stochasticity, e.g. noise or natural variation. Class labels can be inferred by learning to reverse the process: predicting $z$ from $x$, and $y$ from $z$, integrating over all $z$: $p_{\theta,\phi}(y|x) = \int_z p_\theta(y|z)q_\phi(z|x)$.[3] It is generally intractable to learn parameters $(\theta, \phi)$ of this predictive model by maximising the log likelihood, $\int_{x,y} p(x,y) \log p_{\theta,\phi}(y|x)$. Instead a lower bound on the log likelihood can be maximised, comparable to the evidence lower bound (ELBO) used to train a variational auto-encoder (VAE) (Kingma & Welling, 2014; Rezende et al., 2014).

We show that training a softmax classifier under cross entropy loss (SCE) is, in fact, a special case of training this generalised latent classification model under the variational objective, in which the input to the softmax layer ($z$ of Eqn. 1) is treated as the latent variable, the *encoder* parameterises $q_\phi(z|x)$, and the softmax layer computes $p_\theta(y|z)$. In other words, the latent variable model and its training objective provide an *interpretable generalisation of softmax classification*. Probing further, the softmax layer can be interpreted as applying Bayes' rule, $p_\theta(y|z) = \frac{p_\theta(z|y)p_\theta(y)}{\sum_{y'} p_\theta(z|y')p_\theta(y')}$, assuming that latent variables follow *exponential family* class-conditional distributions $p_\theta(z|y)$ for true class distributions to be output. Meanwhile, the distribution that latents *actually* follow, $q_\phi(z|y) = \int_x q_\phi(z|x)p(x|y)$, is defined by the data distribution and the encoder. We refer to these two descriptions of $p(z|y)$ as the *anticipated* and *empirical* latent distributions, respectively, and consider their relationship. We show, both theoretically and empirically, that in practical settings these distributions can materially differ. Indeed, optimising the SCE objective may cause each empirical distribution $q_\phi(z|y)$ to *collapse to a point* rather than *fit* the anticipated $p_\theta(z|y)$. This essentially overfits to the data and loses information required for estimating confidence or other potential downstream tasks, limiting the use of $z$ as a *representation* of $x$. To address the potential discrepancy between $q_\phi(z|y)$ and $p_\theta(z|y)$, so that the softmax layer receives the distribution it expects, we minimise the Kullback-Leibler (KL) divergence between them. This is non-trivial since $q_\phi(z|y)$ can only be sampled not evaluated, hence we use the *density ratio trick* (Nguyen et al., 2010; Gutmann & Hyvärinen, 2010), as seen elsewhere (Makhzani et al., 2015; Mescheder et al., 2017), to approximate the required log probability ratios as an auxiliary task.

The resulting *Variational Classification* (**VC**) objective generalises softmax cross-entropy classification from a latent perspective and fits empirical latent distributions $q_\phi(z|y)$ to anticipated *class priors* $p_\theta(z|y)$. Within this more interpretable framework, latent variables learned by a typical softmax classifier can be considered *maximum likelihood* (MLE) point estimates that *maximise* $p_\theta(y|z)$. By comparison, the two KL components introduced in variational classification, respectively lead to *maximum a posteriori* (MAP) point estimates; and a *Bayesian* treatment where latent variables (approximately) *fit* the full distribution $p_\theta(z|y)$ (Figure 1).[4] Since Variational Classification serves to mitigate over-fitting, which naturally reduces with increased samples, VC is anticipated to offer greatest benefit in low data regimes.

---

[3]We use the notation $q_\phi$ to distinguish distributions, as will be made clear.
[4]Terms of the standard ELBO can be interpreted similarly.

Through a series of experiments on vision and text datasets, we demonstrate that VC achieves comparable accuracy to regular softmax classification while the aligned latent distribution improves calibration, robustness to adversarial perturbations (specifically FGSM "white box"), generalisation under domain shift and performance in low data regimes. Although many prior works target any *one* of these pitfalls of softmax classification, often requiring extra hyperparameters to be tuned on held-out validation sets, VC *simultaneously improves them all*, without being tailored towards any or needing further hyperparameters or validation data. Overall, the VC framework gives novel mathematical insight and interpretability to softmax classification: the encoder maps a mixture of unknown data distributions $p(x|y)$ to a mixture of chosen latent distributions $p_\theta(z|y)$, which the softmax/output layer "flips" by Bayes' rule. This understanding may enable principled improvement of classification and its integration with other latent variable paradigms (e.g. VAEs).

## 2 Background

The proposed generalisation from softmax to variational classification (§3) is analogous to how a deterministic auto-encoder relates to a *variational auto-encoder* (VAE), as briefly summarised below.

Estimating parameters of a latent variable model of the data $p_\theta(x) = \int_z p_\theta(x|\mathrm{z})p_\theta(\mathrm{z})$ by maximising the likelihood, $\int_x p(x) \log p_\theta(x)$, is often intractable. Instead, one can maximise the *evidence lower bound* (ELBO):

$$\int_x p(x) \log p_\theta(x) = \int_x p(x) \int_z q_\phi(z|x) \Big\{ \log p_\theta(x|z) - \log \tfrac{q_\phi(z|x)}{p_\theta(z)} + \log \tfrac{q_\phi(z|x)}{p_\theta(z|x)} \Big\}$$

$$\geq \int_x p(x) \int_z q_\phi(z|x) \Big\{ \log p_\theta(x|z) - \log \tfrac{q_\phi(z|x)}{p_\theta(z)} \Big\} \doteq \textbf{ELBO}, \qquad (2)$$

where $q_\phi(z|x)$ is the *approximate posterior* and the term dropped in the inequality is a Kullback-Leibler (KL) divergence, $D_{\mathrm{KL}}[q(z) \| p(z)] \doteq \int_z q(z) \log \tfrac{q(z)}{p(z)} \geq 0$. The VAE (Kingma & Welling, 2014; Rezende et al., 2014) uses the ELBO as a training objective with $p_\theta(x|z)$ and $q_\phi(z|x)$ assumed to be Gaussian parameterised by neural networks. Setting the variance of $q_\phi(\mathrm{z}|x)$ to zero, i.e. each $q_\phi(\mathrm{z}|x)$ to a delta distribution, the first ("reconstruction") term of Eqn. 2 equates to the training objective of a deterministic *auto-encoder*, which the VAE can be interpreted to probabilistically generalise, allowing for uncertainty or stochasticity in $q_\phi(\mathrm{z}|x)$ constrained by the second ("regularisation") term.

Maximising the ELBO directly equates to minimising $D_{\mathrm{KL}}[p(x) \| p_\theta(x)] + \mathbb{E}_x[D_{\mathrm{KL}}[q_\phi(z|x) \| p_\theta(z|x)]]$, and so fits the model $p_\theta(x)$ to the data distribution $p(x)$ and $q_\phi(z|x)$ to the model posterior $p_\theta(z|x) \doteq \frac{p_\theta(x|z)p_\theta(z)}{p_\theta(x)}$. Equivalently, the modelled distributions $q_\phi(z|x)$ and $p_\theta(x|z)$ are made *consistent under Bayes' rule*.

## 3 Variational Classification

**Classification Latent Variable Model (LVM)**: Consider data $x \in \mathcal{X}$ and labels $y \in \mathcal{Y}$ as samples of random variables x, y jointly distributed $p(\mathrm{x}, \mathrm{y})$. Under the (Markov) generative model in Figure 2 (*left*),

$$p(x) = \int_{y,z} p(x|z)p(z|y)p(y) , \qquad (3)$$

labels can be predicted by reversing the process,

$$p_\theta(y|x) = \int_z p_\theta(y|z)p_\theta(z|x) . \qquad (4)$$

A neural network (NN) softmax classifier is a deterministic function that maps each data point $x$, via a sequence of intermediate representations, to a point on

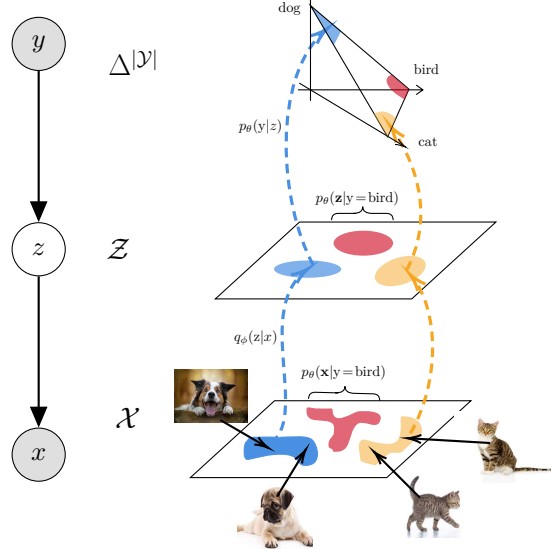

Figure 2: Variational Classification, reversing the generative process: $q_\phi(\mathrm{z}|x)$ maps data $x \in \mathcal{X}$ to the latent space $\mathcal{Z}$, where *empirical* distributions $q_\phi(\mathrm{z}|y)$ are fitted to *class priors* $p_\theta(\mathrm{z}|y)$; top layer computes $p_\theta(y|z)$ by Bayes' rule to give a class prediction $p(y|x)$.

the simplex $\Delta^{|\mathcal{Y}|}$ that parameterises a categorical label distribution $p_\theta(\mathrm{y}|x)$. Any intermediate representation $z = g(x)$ can be considered a sample of a *latent* random variable z from conditional distribution $p(\mathrm{z}|x) = \delta_{z-g(x)}$.

**Proposition**: a NN softmax classifier is a special case of Eqn. 4.

**Proof**: Define (i) the input to the softmax layer as latent variable z; (ii) $p_\theta(z|x) = \delta_{z-f_\omega(x)}$, a delta distribution parameterised by $f_\omega$, the NN up to the softmax layer (the *encoder*); and (iii) $p_\theta(\mathrm{y}|z)$ by the softmax layer (as defined in RHS of Eqn. 1).

### 3.1 Training a Classification LVM

Similarly to the latent variable model for $p_\theta(\mathrm{x})$ (§2), parameters of Eqn. 4 cannot in general be learned by directly maximising the likelihood. Instead we can maximise a lower bound:

$$
\begin{aligned}
\int_{x,y} p(x,y) \log p_\theta(y|x) &= \int_{x,y} p(x,y) \int_z q_\phi(z|x) \Big\{ \log p_\theta(y|z,\cancel{x}) - \cancel{\log \tfrac{q_\phi(z|x)}{p_\theta(z|x)}} + \log \tfrac{q_\phi(z|x)}{p_\theta(z|x,y)} \Big\} \\
&\geq \int_{x,y} p(x,y) \int_z q_\phi(z|x) \log p_\theta(y|z) \;\doteq\; \mathbf{ELBO_{VC}}
\end{aligned}
\tag{5}
$$

Here, $p_\theta(y|z,x) = p_\theta(y|z)$ by the Markov model, and the (freely chosen) variational posterior $q_\phi$ is assumed to depend only on $x$ and set equal to $p_\theta(z|x)$ (eliminating the second term).[5] The derivation of Eqn. 5 follows analogously to that of Eqn. 2 conditioned on $x$; an alternative derivation follows from Jensen's inequality.

Unlike for the standard ELBO, the "dropped" KL term $D_{\mathrm{KL}}[\, q_\phi(z|x) \| p_\theta(z|x,y)]$ (minimised implicitly as $\mathrm{ELBO_{VC}}$ is maximised) may not minimise to zero – except in the limiting case $p_\theta(y|x,z) = p_\theta(y|z)$. That is, when z is a *sufficient statistic* for y given x, intuitively meaning that z contains all information contained in x about y.[6] Hence, maximising $\mathrm{ELBO_{VC}}$ implicitly encourages $z$ to learn a sufficient statistic for $y|x$.

**Proposition**: softmax cross-entropy (SCE) loss is a special case of $\mathrm{ELBO_{VC}}$.

**Proof**: In Eqn. 5, let (i) $q_\phi(z|x) = \delta_{z-f_\omega(x)}$; and (ii) $p_\theta(z|y) = h(z) \exp\{z^\top w_y + b'_y\}, \forall y \in \mathcal{Y}$, for constants $w_y, b'_y$, arbitrary positive function $h : \mathcal{Z} \to \mathbb{R}^+$ and $b_y = b'_y + \log p_\theta(y)$:

$$
\begin{aligned}
\int_{x,y} p(x,y) \int_z q_\phi(z|x) \log p_\theta(y|z) &\overset{(i)}{=} \int_{x,y} p(x,y) \log p_\theta(y|z = f_\omega(x)) \overset{(Bayes)}{=} \int_{x,y} p(x,y) \log \frac{p_\theta(z=f_\omega(x)|y) p_\theta(y)}{\sum_{y'} p_\theta(z=f_\omega(x)|y') p_\theta(y')} \\
&\overset{(ii)}{=} \int_{x,y} p(x,y) \log \frac{\cancel{h(z)} \exp\{f_\omega(x)^\top w_y + b_y\}}{\sum_{y'} \cancel{h(z)} \exp\{f_\omega(x)^\top w_{y'} + b_{y'}\}} \doteq \mathbf{SCE}.
\end{aligned}
\tag{6}
$$

**Corollary**: A NN softmax classifier outputs true label distributions $p(y|x)$ if inputs to the softmax layer, $z$, follow **anticipated** class-conditional distributions $p_\theta(z|y)$ of *(equi-scale) exponential family* form.

### 3.2 Anticipated vs Empirical Latent Distributions

Defining an LVM for classification (Eqn. 4) requires specifying $p_\theta(y|z)$. In the special case of softmax classification, $p_\theta(y|z)$ is effectively encoded by Bayes' rule assuming exponential family $p_\theta(z|y)$, i.e. distributions over softmax layer inputs for class $y$ (Eqn. 6). More generally, one can choose the parametric form of $p_\theta(z|y)$ and compute $p_\theta(y|z)$ by Bayes' rule in a classifier's *output layer* (generalising the standard softmax layer), thereby encoding the distribution latent variables are *anticipated* to follow for accurate label predictions $p(y|x)$ to be output. A natural question then is: *do latent variables of a classification LVM **empirically** follow the **anticipated** distributions $p_\theta(z|y)$?*

Empirical latent distributions are not fixed, but rather defined by $q_\phi(z|y) \doteq \int_x q_\phi(z|x) p(x|y)$, i.e. by sampling $q_\phi(z|x)$ (parameterised by the encoder $f_\omega$) given class samples $x \sim p(\mathrm{x}|y)$. Since $\mathrm{ELBO_{VC}}$ is optimised w.r.t. parameters $\phi$, if optimal parameters are denoted $\phi^*$, the question becomes: *does $q_{\phi^*}(z|y) = p_\theta(z|y)$?*

It can be seen that $\mathrm{ELBO_{VC}}$ is optimised w.r.t $\phi$ if $q_{\phi^*}(z|x) = \delta_{z-z_x}$, for $z_x = \arg\max_z \mathbb{E}_{y|x}[\log p_\theta(y|z)]$ (see appendix A.1).[7] In practice, *true* label distributions $p(y|x)$ are unknown and we have only finite samples

---

[5]We use the notation "$q_\phi$" by analogy to the VAE and to later distinguish $q_\phi(z|y)$, derived from $q_\phi(z|x)$, from $p_\theta(z|y)$.

[6]Proof: from $p(z|x,y) p(y|x) = p(y|x,z) p(z|x)$ and Markovianity, we see that $D_{\mathrm{KL}}[\, q_\phi(z|x) \| p_\theta(z|x,y)] = 0 \Leftrightarrow p_\theta(z|x,y) = q_\phi(z|x) \Leftrightarrow p_\theta(y|x) = p_\theta(y|x,z) = p_\theta(y|z) \Leftrightarrow$ z a sufficient statistic for y|x.

[7]We assume the parametric family $q_\phi$ is sufficiently flexible to closely approximate the analytic maximiser of $\mathrm{ELBO_{VC}}$.

from them. For a continuous data domain $\mathcal{X}$, e.g. images or sounds, any empirically observed $x$ is sampled twice with probability zero and so *is observed once with a single label $y(x)$*. A similar situation arises (for any $\mathcal{X}$) if – as a property of the data – every $x$ has only one ground truth label $y(x)$, i.e. labels are mutually exclusive and *partition* the data.[8] In either case, the expectation over labels simplifies and, for a given class $y$, $z_x = \arg\max_z p_\theta(y|z)$, meaning the optimal latent distribution $q_{\phi^*}(z|x)$ is identical for all samples $x$ of class $y$.[9] Letting $z_y$ denote the optimal latent variable for all $x$ of class $y$, optimal class-level distributions are simply $q_{\phi^*}(z|y) = \delta_{z-z_y}$, and **ELBO$_{\mathbf{VC}}$ is maximised if all latent representations of a class, and hence $q_\phi(z|y)$, "collapse" to the same point**, irrespective of the anticipated $p_\theta(z|y)$.

Since softmax classification is a special case, this reveals the potential for softmax classifiers to learn over-concentrated, or *over-confident*, latent distributions relative to anticipated distributions (subject to the data distribution and model flexibility). In practical terms, the softmax cross-entropy loss may be minimised when all samples of a given class are mapped (by the encoder $f_\omega$) to the same latent variable/representation, regardless of differences in the samples' probabilities or semantics, thus disregarding information that may be useful for calibration or downstream tasks. We note that the Information Bottleneck Theory (Tishby et al., 2000; Tishby & Zaslavsky, 2015; Shwartz-Ziv & Tishby, 2017) assumes that such "loss of information" is beneficial, but as we see below, it is *unnecessary* for classification and may be undesirable in general.

### 3.2.1 Aligning the Anticipated and Empirical Latent Distributions

We have shown that the ELBO$_{\mathrm{VC}}$ objective, a generalisation of SCE loss, effectively involves two versions of the latent class conditional distributions, $p_\theta(z|y)$ and $q_\phi(z|y)$, and that a mismatch between them may have undesirable consequences in terms of information loss. We therefore propose to align $p_\theta(z|y)$ and $q_\phi(z|y)$, or, equivalently, for $p_\theta(y|z)$ and $q_\phi(z|y)$ to be made *consistent under Bayes' rule* (analogous to $p_\theta(x|z)$ and $q_\phi(z|x)$ in the ELBO, §2). Specifically, we minimise $D_{\mathrm{KL}}[q_\phi(z|y)\|p_\theta(z|y)]$, $\forall y \in \mathcal{Y}$. Including this constraint (weighted by $\beta > 0$) and learning required class distribution $p_\pi(\mathrm{y})$ defines the full **VC objective**:

$$-\mathcal{L}_{\mathbf{VC}} = \int_{x,y} p(x,y) \left\{ \int_z q_\phi(z|x) \log \frac{p_\theta(z|y)p_\pi(y)}{\sum_{y'} p_\theta(z|y')p_\pi(y')} - \beta \int_z q_\phi(z|y) \log \frac{q_\phi(z|y)}{p_\theta(z|y)} + \log p_\pi(y) \right\}. \quad (7)$$

Taken incrementally, $q_\phi$–terms of $\mathcal{L}_{\mathbf{VC}}$ can be interpreted as treating the latent variable z from a *maximum likelihood* (MLE), *maximum a posteriori* (MAP) and *Bayesian* perspective:

(i) maximising $\int_z q_\phi(z|x) \log p_\theta(y|z)$ may overfit $q_\phi(z|y) \approx \delta_{z-z_y}$ (as above);  [MLE]

(ii) adding *class priors* $\int_z q_\phi(z|y) \log p_\theta(z|y)$ changes the point estimates $z_y$;  [MAP]

(iii) adding *entropy* $= -\int_z q_\phi(z|y) \log q_\phi(z|y)$ encourages $q_\phi(\mathrm{z}|y)$ to "fill out" $p_\theta(\mathrm{z}|y)$.  [Bayesian]

Figure 1 shows samples from empirical latent distributions $q_\phi(\mathrm{z}|y)$ for classifiers trained under incremental terms of the VC objective. This empirically confirms that softmax cross-entropy loss does not impose the anticipated latent distribution encoded in the output layer (*left*). Adding class priors $p_\theta(\mathrm{z}|y)$ changes the point at which latents of a class concentrate (*centre*). Adding entropy encourages class priors to be "filled out" (*right*), relative to previous point estimates/$\delta$-distributions. As above, if each $x$ has a single label (e.g. MNIST), the MLE/MAP training objectives are optimised when class distributions $q_\phi(\mathrm{z}|y)$ collapse to a point. We note that complete collapse is not observed in practice (Figure 1, *left*, *centre*), which we conjecture is due to strong constraints on $f_\omega$, in particular continuity and $\ell_2$ regularisation and early stopping based on validation loss. Compared to the KL form of the ELBO (§2), maximising Eqn. 7 is equivalent to minimising:

$$\underline{\mathbb{E}_x \left[ D_{\mathrm{KL}}[p(y|x)\|p_\theta(y|x)] \right.} + \mathbb{E}_{x,y}\left[ D_{\mathrm{KL}}[q_\phi(z|x)\|p_\theta(z|x,y)] \right] + \mathbb{E}_y\left[ D_{\mathrm{KL}}[q_\phi(z|y)\|p_\theta(z|y)] \right] + D_{\mathrm{KL}}[p(y)\|p_\pi(y)] \quad (8)$$

showing the extra constraints over the core objective of modelling $p(y|x)$ by $p_\theta(y|x)$ (underlined).

---

[8]As in popular image datasets, e.g. MNIST, CIFAR, ImageNet, where *samples belong to one class or another*.

[9]Subject to uniqueness of $\arg\max_z p_\theta(y|z)$, which is not guaranteed in general, but is assumed for suitable $p_\theta(z|y)$, such as the softmax case of central interest: if all $x$ have a single label $y(x)$ (i.e. $p(y|x) = \mathbf{1}_{y=y(x)}$ is a "one-hot" vector), and norms are finitely constrained ($\|z\| = \alpha > 0$), then the SCE objective (Eqn. 6) is maximised, and softmax outputs $p_\theta(y|x)$ (Eqn. 1) increasingly approximate true $p(y|x)$, as class parameters $w_y$ are maximally dispersed (i.e. unit vectors $\hat{w}_y$ tend to a regular polytope on the unit sphere) and all representations of a class $y$ align with the class parameter: $z_x = f_\omega(x) \to \alpha\hat{w}_{y(x)}$ (unique).

---

**Algorithm 1** Variational Classification (VC)

---

1: Input      $p_\theta(z|y)$, $q_\phi(z|x)$, $p_\pi(y)$, $T_\psi(z)$; learning rate schedule $\{\eta_\theta^t, \eta_\phi^t, \eta_\pi^t, \eta_\psi^t\}_t$, $\beta$

2: Initialise   $\theta, \phi, \pi, \psi$;   $t \leftarrow 0$

3: **while** not converged **do**

4:     $\{x_i, y_i\}_{i=1}^m \sim \mathcal{D}$                                      [sample batch from data distribution $p(x, y)$]

5:     **for** $z = \{1 \ldots m\}$ **do**

6:        $z_i \sim q_\phi(z|x_i)$, $z_i' \sim p_\theta(z|y_i)$                       [e.g. $q_\phi(z|x_i) \doteq \delta_{z - f_\omega(x_i)}$, $\phi \doteq \omega \Rightarrow z_i = f_\omega(x_i)$]

7:        $p_\theta(y_i|z_i) = \frac{p_\theta(z_i|y_i) p_\pi(y_i)}{\sum_y p_\theta(z_i|y) p_\pi(y)}$

8:     **end for**

9:     $g_\theta \leftarrow \frac{1}{m} \sum_{i=1}^m \nabla_\theta \left[ \log p_\theta(y_i|z_i) + \beta\, p_\theta(z_i|y_i) \right]$

10:    $g_\phi \leftarrow \frac{1}{m} \sum_{i=1}^m \nabla_\phi \left[ \log p_\theta(y_i|z_i) - \beta\, T_\psi(z_i) \right]$                 [e.g. using "reparameterisation trick"]

11:    $g_\pi \leftarrow \frac{1}{m} \sum_{i=1}^m \nabla_\pi \log p_\pi(y_i)$

12:    $g_\psi \leftarrow \frac{1}{m} \sum_{i=1}^m \nabla_\psi \left[ \log \sigma(T_\psi(z_i)) + \log(1 - \sigma(T_\psi(z_i'))) \right]$

13:    $\theta \leftarrow \theta + \eta_\theta^t g_\theta$,     $\phi \leftarrow \phi + \eta_\phi^t g_\phi$,     $\pi \leftarrow \pi + \eta_\pi^t g_\pi$,     $\psi \leftarrow \psi + \eta_\psi^t g_\psi$,      $t \leftarrow t + 1$

14: **end while**

---

### 3.3 Optimising the VC Objective

The VC objective (Eqn. 7) is a lower bound that can be maximised by gradient methods, e.g. SGD:

- the first term can be calculated by sampling $q_\phi(z|x)$ (using the "reparameterisation trick" as necessary (Kingma & Welling, 2014)) and computing $p_\theta(y|z)$ by Bayes' rule;

- the third term is standard multinomial cross-entropy;

- the second term, however, is not readily computable since $q_\phi(z|y)$ is implicit and cannot easily be evaluated, only sampled, as $z \sim q_\phi(z|x)$ (parameterised by $f_\omega$) for class samples $x \sim p(x|y)$.

Fortunately, we require log ratios $\log \frac{q_\phi(z|y)}{p_\theta(z|y)}$ for each class $y$, which can be approximated by training a binary classifiers to distinguish samples of $q_\phi(z|y)$ from those of $p_\theta(z|y)$. This so-called *density ratio trick* underpins learning methods such as Noise Contrastive Estimation (Gutmann & Hyvärinen, 2010) and contrastive self-supervised learning (e.g. Oord et al., 2018; Chen et al., 2020) and has been used comparably to train variants of the VAE (Makhzani et al., 2015; Mescheder et al., 2017).

Specifically, we maximise the following *auxiliary objective* w.r.t. parameters $\psi$ of a set of binary classifiers:

$$-\mathcal{L}_{\mathbf{aux}} = \int_y p(y) \Big\{ \int_z q_\phi(z|y) \log \sigma(T_\psi^y(z)) + \int_z p_\theta(z|y) \log(1 - \sigma(T_\psi^y(z))) \Big\} \tag{9}$$

where $\sigma$ is the logistic sigmoid function $\sigma(x) = (1 + e^{-x})^{-1}$, $T_\psi^y(z) = w_y^\top z + b_y$ and $\psi = \{w_y, b_y\}_{y \in \mathcal{Y}}$.

It is easy to show that Eqn. 9 is optimised if $T_\psi^y(z) = \log \frac{q_\phi(z|y)}{p_\theta(z|y)}$, $\forall y \in \mathcal{Y}$. Hence, when all binary classifiers are trained, $T_\psi^y(z)$ approximates the log ratio for class $y$ required by the VC objective (Eqn. 7). Optimising the VC objective might, in principle, also require gradients of the approximated log ratios w.r.t. parameters $\theta$ and $\phi$. However, the gradient w.r.t. the $\phi$ found within the log ratio is always zero (Mescheder et al., 2017) and so the gradient w.r.t. $\theta$ can be computed from Eqn. 7. See Algorithm 1 for a summary.

This approach is *adversarial* since (a) the VC objective is maximised when log ratios give a *minimal* KL divergence, i.e. when $q_\phi(z|y) = p_\theta(z|y)$ and latents sampled from $q_\phi(z|y)$ or $p_\theta(z|y)$ are indistinguishable; whereas (b) the auxiliary objective is maximised if the ratios are *maximal* and the two distributions are fully discriminated. Relating to a Generative Adversarial Network (GAN) (Goodfellow et al., 2014a), the encoder $f_\omega$ acts as a *generator* and each binary classifier as a *discriminator*. Unlike a GAN, VC requires a discriminator *per class* that each distinguish generated samples from a learned, rather than static, reference/noise distribution $p_\theta(z|y)$. However, whereas a GAN discriminator distinguishes between complex distributions in the data domain, a VC discriminator compares a Gaussian to an approximate Gaussian in the lower dimensional latent domain, a far simpler task. The auxiliary objective does not change the complexity relative to softmax classification and can be parallelised across classes, adding marginal computational overhead per class.

### 3.3.1 Optimum of the VC Objective

In §3.2, we showed that the empirical distribution $q_\phi(z|x)$ that opitimises the $\text{ELBO}_{\text{VC}}$ need not match the anticipated $p_\theta(z|y)$. Here, we perform similar analysis to identify $q_{\phi^*}(z|x)$ that maximises the VC objective, which, by construction of the objective, is expected to better match the anticipated distribution.

Letting $\beta = 1$ to simplify (see appendix A.2 for general case), the VC objective is maximised w.r.t. $q_\phi(z|x)$ if:

$$\mathbb{E}_{p(y|x)}[\log q_\phi(z|y)] = \mathbb{E}_{p(y|x)}[\log p_\theta(y|z)p_\theta(z|y)] + c \ , \tag{10}$$

for a constant $c$. This is satisfied if, for each class $y$,

$$q_\phi(z|y) = p_\theta(z|y)\frac{p_\theta(y|z)}{\mathbb{E}_{p_\theta(z'|y)}[p_\theta(y|z')]} \ , \tag{11}$$

giving a unique solution if each $x$ has a single label $y$ (see §3.2; see appendix A.2 for proof). This shows that each $q_\phi(z|y)$ fits $p_\theta(z|y)$ scaled by a ratio of $p_\theta(y|z)$ to its weighted average. Hence, where $p_\theta(y|z)$ is *above average*, $q_\phi(z|y) > p_\theta(z|y)$, and vice versa. In simple terms, $q_\phi(z|y)$ reflects $p_\theta(z|y)$ but is "peakier" (fitting observation in Figure 1). We have thus shown empirically (Figure 1) and theoretically that the VC objective aligns the empirical and anticipated latent distributions. However, these distributions are not identical and we leave to future work the derivation of an objective that achieves both $p_\theta(y|x) = p(y|x)$ and $q_\phi(z|y) = p_\theta(z|y)$.

### 3.4 Summary

The latent variable model for classification (Eqn. 4) abstracts a typical softmax classifier, giving interpretability to its components:

- the encoder ($f_\omega$) transforms a mixture of analytically unknown class-conditional data distributions $p(\text{x}|y)$ to a mixture of analytically defined latent distributions $p_\theta(z|y)$;

- assuming latent variables follow the anticipated class distributions $p_\theta(\text{z}|\text{y})$, the output layer applies Bayes' rule to give $p_\theta(\text{y}|\text{z})$ (see figure 2) and thus meaningful estimates of label distributions $p(\text{y}|x)$ (by Eqn. 4).

$\text{ELBO}_{\text{VC}}$ generalises softmax cross-entropy, treating the input to the softmax layer as a latent variable and identifying the anticipated class-conditionals $p_\theta(z|y)$ implicitly encoded within the softmax layer. Extending this, the VC objective ($\mathcal{L}_{\text{VC}}$) encourages the empirical latent distributions $q_\phi(z|y)$ to fit $p_\theta(z|y)$. Softmax cross-entropy loss is recovered from $\mathcal{L}_{\text{VC}}$ by setting (i) $q_\phi(\text{z}|x) = \delta_{z - f_\omega(x)}$; (ii) $p_\theta(\text{z}|\text{y})$ to (equal-scale) exponential family distributions, e.g. equivariate Gaussians; and (iii) $\beta = 0$. This is analogous to how a deterministic auto-encoder relates to a VAE. Thus **the VC framework elucidates assumptions made implicitly in softmax classification** and by generalising this special case, allows these assumptions, e.g. the choice of $p_\theta(\text{z}|\text{y})$, to be revised on a task/data-specific basis.

## 4 Related Work

Despite notable differences, the *energy-based* interpretation of softmax classification of Grathwohl et al. (2019) is perhaps most comparable to our own in taking an abstract view to improve softmax classification. However, their gains, e.g. in calibration and adversarial robustness, come at a significant cost to the main aim: classification accuracy. Further, the required MCMC normalisation reportedly slows and destabilises training. In contrast, we use tractable probability distributions and retain the order of complexity. Our approach is also notionally related to Bayesian Neural Networks (BNNs) or related approaches such as MC-dropout (Gal & Ghahramani, 2016), although these are *Bayesian* with respect to model parameters, rather than latent variables. In principle, these mightt be combined (e.g. Murphy, 2012) as an interesting future direction.

Several previous works adapt the standard ELBO, used to learn a model of $p(x)$, to a conditional analog for learning $p(y|x)$ (Tang & Salakhutdinov, 2013; Sohn et al., 2015). However, such works focus on generative scenarios rather than discriminative classification, e.g. $x$ being a face image and $y|x$ being the same face in a different pose determined by latent $z$; or $x$ being part of an image and $y|x$ its completion given latent content $z$. The *Gaussian stochastic neural network* (GSNN) model (Sohn et al., 2015) is closer to our own by conditioning $q(z|x, y)$ only on $x$, however the model neither generalises softmax classification nor considers class-level latent priors $q(z|y)$ as in variational classification.

Variational classification subsumes a number of works that add a regularisation term to a softmax cross-entropy loss function, which can be interpreted as a prior over latent variables in the "MAP" case (§3.2.1). For example, several semi-supervised learning models can be interpreted as treating the softmax *outputs* as latent variables and using a latent prior to guide predictions for unlabelled data (Allen et al., 2020). Closer to variational classification, several works can be interpreted as treating softmax *inputs* as latent variables with a regularisation term that encourages prior beliefs, such as *deterministic* label predictions (i.e. all probability mass on a single class), which can be encouraged by imposing a *large margin* between class-conditional latent distributions (Liu et al., 2016; Wen et al., 2016; Wan et al., 2018; 2022; Scott et al., 2021).

Variational classification also relates to works across several learning paradigms in which a Gaussian mixture prior is imposed in the latent space, e.g. for representation learning (Xie et al., 2016; Caron et al., 2018), in auto-encoders (Song et al., 2013; Ghosh et al., 2019) and in variational auto-encoders (Jiang et al., 2016; Yang et al., 2017; Prasad et al., 2020; Manduchi et al., 2021).

## 5 Empirical Validation

Our goal is to empirically demonstrate that the latent structure induced by the VC objective is beneficial relative to the standard softmax classifier. A variational classifier can be substituted wherever a softmax classifier is used, by making distributional choices appropriate for the data. In particular, variational classification does not set out to address any one drawback of a softmax classifier, rather it aims to better reverse the generative process and so capture the data distribution, providing multiple benefits. We illustrate the effectiveness of a VC through a variety of tasks on familiar datasets from the visual and text domains.

Specifically, we set out to validate the following hypotheses:

**H1:** The VC objective improves uncertainty estimation, leading to a more calibrated model.

**H2:** The VC objective increases model robustness to changes in the data distribution.

**H3:** The VC objective enhances resistance to adversarial perturbations.

**H4:** The VC objective aids learning from fewer samples.

For fair comparison, we make minimal changes to adapt a standard softmax classifier to a variational classifier. As described in §3.4, we train with the VC objective (Eqn. 7) under the following assumptions: $q_\phi(\mathbf{z}|x)$ is a delta distribution parameterised by a neural network $f_\omega : \mathcal{X} \to \mathcal{Z}$; class-conditional priors $p_\theta(\mathbf{z}|y)$ are multi-variate Gaussians with parameters learned from the data (we use diagonal covariance for simplicity). To provide an ablation across the components of the VC objective, we compare classifiers trained to maximise three objective functions (see §3):

**CE:** equivalent to standard softmax cross-entropy under the above assumptions and corresponds to the MLE form of the VC objective (§3.2.1, (i)).

$$J_{\mathbf{CE}} = \int_{x,y} p(x,y) \left( \int_z q_\phi(z|x) \log p_\theta(y|z) + \log p_\pi(y) \right)$$

**GM:** includes class priors and corresponds to the MAP form of the VC objective (§3.2.1, (ii)). This is equivalent to Wan et al. (2018) with just the Gaussian Prior.

$$J_{\mathbf{GM}} = J_{\mathbf{CE}} + \int_{x,y} p(x,y) \int_z q_\phi(z|y) \log p_\theta(z|y)$$

**VC:** includes entropy of the empirical latent distributions and corresponds to the Bayesian form of the VC objective (§3.2.1, (iii)).

$$J_{\mathbf{VC}} = J_{\mathbf{GM}} - \int_{x,y} p(x,y) \int_z q_\phi(z|y) \log q_\phi(z|y)$$

| | CIFAR-10 | | | | CIFAR-100 | | | | TINY-IMAGENET | | |
|---|---|---|---|---|---|---|---|---|---|---|---|
| | CE | GM$^\diamond$ | VC | vMF$^\star$ | CE | GM$^\diamond$ | VC | vMF$^\star$ | CE | GM$^\diamond$ | VC |
| **Acc.** (%, ↑) | | | | | | | | | | | |
| WRN | $96.2_{\pm 0.1}$ | $95.0_{\pm 0.2}$ | $96.3_{\pm 0.2}$ | - | $80.3_{\pm 0.1}$ | $79.8_{\pm 0.2}$ | $80.3_{\pm 0.1}$ | - | - | - | - |
| RNET | $93.7_{\pm 0.1}$ | $93.0_{\pm 0.1}$ | $93.2_{\pm 0.1}$ | $94.0_{\pm 0.1}$ | $73.2_{\pm 0.1}$ | $74.2_{\pm 0.1}$ | $73.4_{\pm 0.1}$ | $69.94_{\pm 0.2}$ | $59.7_{\pm 0.2}$ | $59.3_{\pm 0.1}$ | $59.3_{\pm 0.1}$ |
| **ECE** (%, ↓) | | | | | | | | | | | |
| WRN | $3.1_{\pm 0.2}$ | $3.5_{\pm 0.3}$ | $\mathbf{2.1}_{\pm 0.2}$ | - | $11.1_{\pm 0.7}$ | $19.6_{\pm 0.4}$ | $\mathbf{4.8}_{\pm 0.3}$ | - | - | - | - |
| RNET | $3.8_{\pm 0.3}$ | $4.1_{\pm 0.2}$ | $\mathbf{3.2}_{\pm 0.2}$ | $5.9_{\pm 0.2}$ | $8.7_{\pm 0.2}$ | $10.5_{\pm 0.2}$ | $\mathbf{5.1}_{\pm 0.2}$ | $7.9_{\pm 0.3}$ | $12.3_{\pm 0.4}$ | $8.75_{\pm 0.2}$ | $\mathbf{7.4}_{\pm 0.5}$ |

Table 1: Classification Accuracy and Expected Calibration Error (mean, std.dev. over 5 runs). Accuracy is comparable between VC and CE across encoder architectures and data sets, while calibration of VC notably improves. $\star$ from Scott et al. (2021), $\diamond$ our implementation of Wan et al. (2018)

## 5.1 Accuracy and Calibration

We first compare the classification accuracy and calibration of each model on three standard benchmarks (CIFAR-10, CIFAR-100, and TINY-IMAGENET), across two standard ResNet model architectures (*WideResNet-28-10* (WRN) and *ResNet-50* (RNET)) (He et al., 2016; Zagoruyko & Komodakis, 2016). Calibration is evaluated in terms of the *Expected Calibration Error* (ECE) (see Appendix C). Table 1 shows that the VC and GM models achieve comparable accuracy to softmax cross entropy (CE), but that the VC model is consistently, significantly more calibrated (**H1**). Unlike approaches such as Platt's scaling (Platt et al., 1999) and temperature scaling (Guo et al., 2017), no *post hoc* calibration is performed requiring additional data or associated hyperparameters tuning.

We also compare MC-Dropout (Gal & Ghahramani, 2016) for CIFAR-10 and CIFAR-100 on *ResNet-50* ($p = 0.2$, averaging over 10 samples). As seen previously (Ovadia et al., 2019), although calibration improves relative to CE (3.3%, 1.4%, resp.), the main goal of classification, prediction accuracy, reduces (92.7%, 70.1%).

## 5.2 Generalization under distribution shift

When used in real-world settings, machine learning models may encounter *distribution shift* relative to the training data. It can be important to know when a model's output is reliable and can be trusted, requiring the model to be **calibrated on out-of-distribution (OOD) data** and *know when they do not know*. To test performance under distribution shift, we use the robustness benchmarks, CIFAR-10-C, CIFAR-100-C and TINY-IMAGENET-C, proposed by Hendrycks & Dietterich (2019), which *simulate* distribution shift by adding various *synthetic* corruptions of varying intensities to a dataset. We compare the CE model, with and without temperature scaling, to the VC model. Temperature scaling was performed as in Guo et al. (2017) with the temperature tuned on an in-distribution validation set.

Both models are found to perform comparably in terms of classification accuracy (Figure 8), according to previous results (§5.1). However, Figure 3 shows that the VC model has a consistently lower calibration error as the corruption intensity increases (left to right) (**H2**). We note that the improvement in calibration between the CE and VC models increases as the complexity of the dataset increases.

When deployed in the wild, *natural* distributional shifts may occur in the data due to subtle changes in the data generation process, e.g. a change of camera. We test resilience to *natural* distributional shifts on two tasks: Natural Language Inference (NLI) and detecting whether cells are cancerous from microscopic

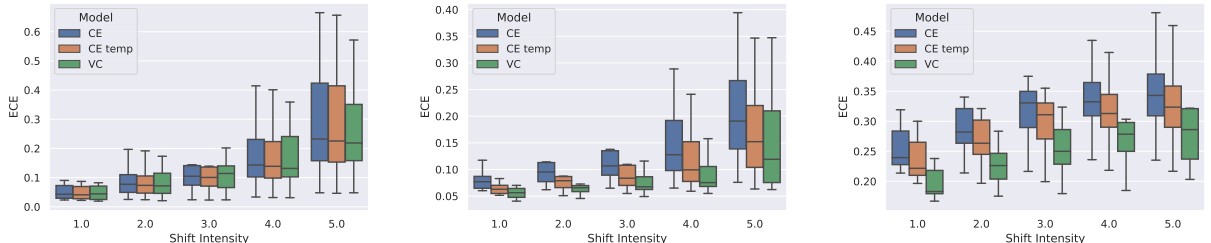

Figure 3: Calibration under distribution shift: (*l*) CIFAR-10-C, (*m*) CIFAR-100-C, (*r*) TINY-IMAGENET-C. Boxes indicate quartiles, whiskers indicate min/max, across 16 types of synthetic distribution shift.

images. NLI requires verifying if a hypothesis logically follows from a premise. Models are trained on the SNLI dataset (Bowman et al., 2015) and tested on the MNLI dataset (Williams et al., 2018) taken from more diverse sources. Cancer detection uses the CAMELYON17 dataset (Bandi et al., 2018) from the WILDs datasets (Koh et al., 2021), where the `train` and `eval` sets contain images from different hospitals.

Table 2 shows that the VC model achieves better calibration under these natural distributional shifts (**H2**). The CAMELYON17 (CAM) dataset has a relatively small number (1000) of training samples (hence wide error bars are expected), which combines distribution shift with a low data setting (**H4**) and shows that the VC model achieves higher (average) accuracy in this more challenging real-world setting.

|  | Accuracy ($\uparrow$) | | Calibration ($\downarrow$) | |
|---|---|---|---|---|
|  | CE | VC | CE | VC |
| NLI | **71.2** ± 0.1 | **71.2** ± 0.1 | 7.3 ± 0.2 | **3.4** ± 0.2 |
| CAM | 79.2 ± 2.8 | **84.5** ± 4.0 | 8.4 ± 2.5 | **1.8** ± 1.3 |

Table 2: Accuracy and Calibration (ECE) under distributional shift (mean, std. err., 5 runs)

We also test the ability to **detect OOD examples**. We compute the AUROC when a model is trained on CIFAR-10 and evaluated on the CIFAR-10 validation set mixed (in turn) with SVHN, CIFAR-100, and CELEBA (Goodfellow et al., 2013; Liu et al., 2015). We compare the VC and CE models using the probability of the predicted class $\arg\max_y p_\theta(y|x)$ as a means of identifying OOD samples.

Table 3 shows that the VC model performs comparably to the CE model. We also consider $p(z)$ as a metric to detect OOD samples and achieve comparable results, which is broadly consistent with the findings of (Grathwohl et al., 2019). Although the VC model learns to map the data to a more structured latent space and, from the results above, makes more calibrated predictions for OOD data, it does not appear to be better able to distinguish OOD data than a standard softmax classifier (CE) using the metrics tested (we note that "OOD" is a loosely defined term).

| Model | SVHN | C-100 | CelebA |
|---|---|---|---|
| $P_{\text{CE}}(y|x)$ | 0.92 | 0.88 | 0.90 |
| $P_{\text{VC}}(y|z)$ | 0.93 | 0.86 | 0.89 |

Table 3: AUROC for OOD detection. Models trained on CIFAR-10, evaluated on in and out-of-distribution samples.

## 5.3 Adversarial Robustness

We test model robustness to adversarially generated images using the common *Fast Gradient Sign Method* (FGSM) of adversarial attack (Goodfellow et al., 2014b). This "attack" is arbitrarily chosen and VC is not explicitly tailored towards it. Perturbations are generated as $P = \epsilon \times sign(\nabla_x \mathcal{L}(x, y))$, where $\mathcal{L}(x, y)$ is the model loss for data sample $x$ and correct class $y$; and $\epsilon$ is the attack *magnitude*. We compare all models trained on MNIST and CIFAR-10 against FGSM attacks of different magnitudes.

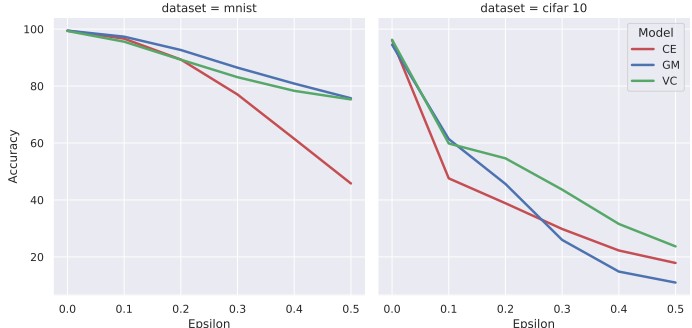

Figure 4: Prediction accuracy for increasing FGSM adversarial attacks *(l)* MNIST; *(r)* CIFAR-10

Results in Figure 4 show that the VC model is consistently more (FGSM) adversarially robust relative to the standard CE model, across attack magnitudes on both datasets (**H3**).

## 5.4 Low Data Regime

In many real-world settings, datasets may have relatively few data samples and it may be prohibitive or impossible to acquire more, e.g. historic data or rare medical cases. We investigate model performance when data is scarce on the hypothesis that a prior over the latent space enables the model to better generalise from fewer samples. Models are trained on 500 samples from MNIST, 1000 samples from CIFAR-10 and 50 samples from AGNEWS.

|  | CE | GM | VC |
|---|---|---|---|
| MNIST | 93.1 ± 0.2 | **94.4** ± 0.1 | **94.2** ± 0.2 |
| CIFAR-10 | 52.7 ± 0.5 | 54.2 ± 0.6 | **56.3** ± 0.6 |
| AGNEWS | 56.3 ± 5.3 | 61.5± 2.9 | **66.3** ± 4.6 |

Table 4: Accuracy in low data regime (mean, std.err., 5 runs)

Results in Table 4 show that introducing the prior (GM) improves performance in a low data regime and that the additional entropy term in the VC model maintains or further improves accuracy (**H4**), particularly on the more complex datasets.

We further probe the relative benefit of the VC model over the CE baseline as training sample size varies (**H4**) on 10 MedMNIST classifcation datasets (Yang et al., 2021), a collection of real-world medical datasets of varying sizes.

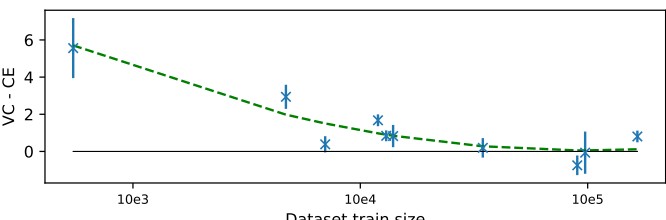

Figure 5 shows the increase in classification accuracy for the VC model relative to the CE model against number of training samples (log scale). The results show a clear trend that the benefit of the additional latent structure imposed in the VC model increases exponentially as the number of training samples decreases.

Figure 5: Accuracy increase of VC vs CE on 10 MedMNIST classification datasets of varying training set size. Blue points indicate accuracy on a dataset (mean, std.err., 3 runs). Green line shows a best-fit trend across dataset size.

Together with the results in Table 4, this suggests that the VC model offers most significant benefit for small, complex datasets.

## 6 Conclusion

We present Variational Classification (VC), a latent generalisation of standard softmax classification trained under cross-entropy loss, mirroring the relationship between the variational auto-encoder and the deterministic auto-encoder (§3). We show that softmax classification is a special case of VC under specific assumptions that are effectively taken for granted when using a softmax output layer. Moreover we see that latent distributional assumptions, "hard-coded" in the softmax layer and anticipated to be followed for accurate class predictions, are neither enforced theoretically nor satisfied empirically. We propose a novel training objective based on the ELBO to better align the *empirical* latent distribution to that *anticipated*. A series of experiments on image and text datasets show that, with marginal computational overhead and without tuning hyper-parameters other than for the original classification task, variational classification achieves comparable prediction accuracy to standard softmax classification while significantly improving calibration, adversarial robustness (specifically FGSM), robustness to distribution shift and performance in low data regimes.

In terms of limitations, we intentionally focus on the *output* layer of a classifier, treating the encoder $f_\omega$ as a "black-box". This leaves open question of how, and how well, the underlying neural network achieves its role of transforming a mixture of unknown data distributions $p(x|y)$ to a mixture of specified latent distributions $p(z|y)$. We also prove that optimal *empirical* latent distributions $q_\phi(z|y)$ are "peaky" approximations to the *anticipated* $p_\theta(z|y)$, leaving open the possibility of further improvement to the VC objective.

The VC framework gives new theoretical insight into the highly familiar softmax classifier, opening up several interesting future directions. For example, $q(z|x)$ might be modelled by a stochastic distribution, rather than a delta distribution, to reflect uncertainty in the latent variables, similarly to a VAE. VC may also be extended to semi-supervised learning and related to approaches that impose structure in the latent space.

## 7 Acknowledgements

Carl is gratefully supported by an ETH AI Centre Postdoctoral Fellowships and a small projects grant from the Haslerstiftung (no. 23072). Mrinmaya acknowledges support from the Swiss National Science Foundation (Project No. 197155), a Responsible AI grant by the Haslerstiftung; and an ETH Grant (ETH-19 21-1).

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

# A Proofs

## A.1 Optimising the ELBO$_{\text{VC}}$ w.r.t $q$

Rearranging Eqn. 5, the ELBO$_{\text{VC}}$ is optimised by

$$
\underset{q_\phi(z|x)}{\arg\max} \int_x \sum_y p(x, y) \int_z q_\phi(z|x) \log p_\theta(y|z)
$$

$$
= \underset{q_\phi(z|x)}{\arg\max} \int_x p(x) \int_z q_\phi(z|x) \sum_y p(y|x) \log p_\theta(y|z)
$$

The integral over $z$ is a $q_\phi(z|x)$-weighted sum of $\sum_y p(y|x) \log p_\theta(y|z)$ terms. Since $q_\phi(z|x)$ is a probability distribution, the integral is upper bounded by $\max_z \sum_y p(y|x) \log p_\theta(y|z)$. This maximum is attained *iff* support of $q_\phi(z|x)$ is restricted to $z^* = \arg\max_z \sum_y p(y|x) \log p_\theta(y|z)$ (which may not be unique). $\qquad \square$

## A.2 Optimising the VC objective w.r.t. $q$

Setting $\beta = 1$ in Eqn. 7 to simplify and adding a lagrangian term to constrain $q_\phi(z|x)$ to a probability distribution, we aim to find

$$
\underset{q_\phi(z|x)}{\arg\max} \int_x \sum_y p(x, y) \Big\{ \int_z q_\phi(z|x) \log p_\theta(y|z)
$$

$$
- \int_z q_\phi(z|y) \log \tfrac{q_\phi(z|y)}{p_\theta(z|y)} + \log p_\pi(y) \Big\} + \lambda \big(1 - \int_z q_\phi(z|x)\big) .
$$

Recalling that $q_\phi(z|y) = \int_x q_\phi(z|x) p(x|y)$ and using calculus of variations, we set the derivative of this functional w.r.t. $q_\phi(z|x)$ to zero

$$
\sum_y p(x, y) \Big\{ \log p_\theta(y|z) - \big(\log \tfrac{q_\phi(z|y)}{p_\theta(z|y)} + 1\big) \Big\} - \lambda = 0
$$

Rearranging and diving through by $p(x)$ gives

$$
\mathbb{E}_{p(y|x)}[\log q_\phi(z|y)] = \mathbb{E}_{p(y|x)}[\log p_\theta(y|z) p_\theta(z|y)] + c ,
$$

where $c = -(1 + \tfrac{\lambda}{p(x)})$. Further, if each label $y$ occurs once with each $x$, due to sampling or otherwise, then this simplifies to

$$
q_\phi(z|y^*) e^c = p_\theta(y^*|z) p_\theta(z|y^*) ,
$$

which holds for all classes $y \in \mathcal{Y}$. Integrating over $z$ shows $e^c = \int_z p_\theta(y|z) p_\theta(z|y)$ to give

$$
q_\phi(z|y) = \frac{p_\theta(y|z) p_\theta(z|y)}{\int_z p_\theta(y|z) p_\theta(z|y)} = p_\theta(z|y) \frac{p_\theta(y|z)}{\mathbb{E}_{p_\theta(z|y)}[p_\theta(y|z)]} . \qquad \square
$$

We note, it is straightforward to include $\beta$ to show

$$
q_\phi(z|y) = p_\theta(z|y) \frac{p_\theta(y|z)^{1/\beta}}{\mathbb{E}_{p_\theta(z|y)}[p_\theta(y|z)^{1/\beta}]} .
$$

## B  Justifying the Latent Prior in Variational Classification

Choosing Gaussian class priors in Variational classification can be interpreted in two ways:

**Well-specified generative model**: Assume data $x \in \mathcal{X}$ is generated from the hierarchical model: $y \to z \to x$, where $p(y)$ is categorical; $p(z|y)$ are analytically known distributions, e.g. $\mathcal{N}(z; \mu_y, \Sigma_y)$; the dimensionality of z is not large; and $x = h(z)$ for an arbitrary invertible function $h : \mathcal{Z} \to \mathcal{X}$ (if $\mathcal{X}$ is of higher dimension than $\mathcal{Z}$, assume $h$ maps one-to-one to a manifold in $\mathcal{X}$). Accordingly, $p(x)$ is a mixture of unknown distributions. If $\{p_\theta(z|y)\}_\theta$ includes the true distribution $p(z|y)$, variational classification effectively aims to invert $h$ and learn the parameters of the true generative model. In practice, the model parameters and $h^{-1}$ may only be identifiable up to some equivalence, but by reflecting the true latent variables, the learned latent variables should be semantically meaningful.

**Miss-specified model**: Assume data is generated as above, but with z having a large, potentially uncountable, dimension with complex dependencies, e.g. details of every blade of grass or strand of hair in an image. In general, it is impossible to learn all such latent variables with a lower dimensional model. The latent variables of a VC might learn a complex function of multiple true latent variables.

The first scenario is ideal since the model might learn disentangled, semantically meaningful features of the data. However, it requires distributions to be well-specified and a low number of true latent variables. For natural data with many latent variables, the second case seems more plausible but choosing $p_\theta(z|y)$ to be Gaussian may nevertheless be justifiable by the Central Limit Theorem.

## C  Calibration Metrics

One way to measure if a model is calibrated is to compute the expected difference between the confidence and expected accuracy of a model.

$$\mathbb{E}_{P(\hat{y}|x)}\Big[\mathbb{P}(\hat{y} = y | P(\hat{y}|x) = p) - p\Big] \tag{12}$$

This is known as expected calibration error (ECE) (Naeini et al., 2015). Practically, ECE is estimated by sorting the predictions by their confidence scores, partitioning the predictions in $M$ equally spaced bins $(B_1 \dots B_M)$ and taking the weighted average of the difference between the average accuracy and average confidence of the bins. In our experiments we use 20 equally spaced bins.

$$\text{ECE} = \sum_{m=1}^{M} \frac{|B_m|}{n} |acc(B_m) - conf(B_m)| \tag{13}$$

## D  OOD Detection

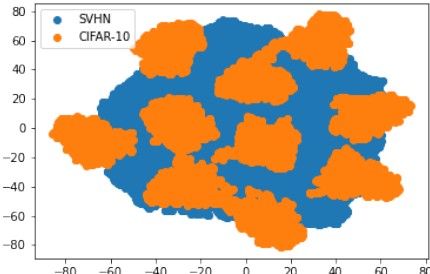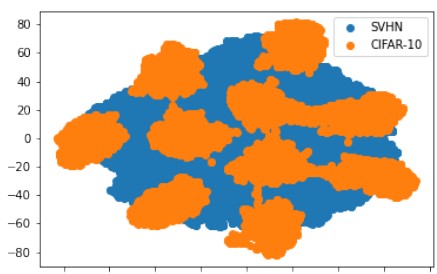

Figure 6: t-SNE plots of the feature space for a classifier trained on CIFAR-10. *(l)* Trained using CE. *(r)* Trained using VC. We posit that similar to CE, VC model is unable to meaningfully represent data from an entirely different distribution.

# E    Semantics of the latent space

To try to understand the semantics captured in the latent space, we use a pre-trained MNIST model on the *Ambiguous MNIST* dataset (Mukhoti et al., 2021). We interpolate between ambiguous 7's that are mapped close to the Gaussian clusters of classes of "1" and "2". It can be observed that traversing from the mean of the "7" Gaussian to that on the "1" class, the ambiguous 7's begin to look more like "1"s.

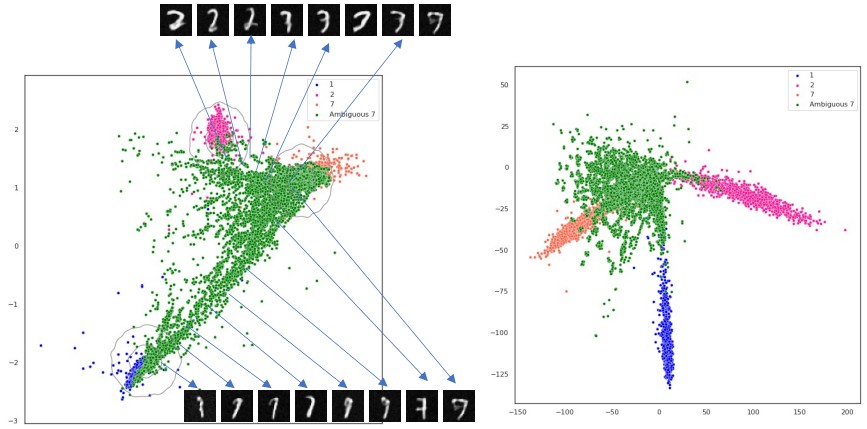

Figure 7: Interpolating in the latent space: Ambiguous MNIST when mapped on the latent space. *(l) VC, (r) CE*

# F    Classification under Domain Shift

A comparison of accuracy between the VC and CE models under 16 different synthetic domain shifts. We find that VC performs comparably well as CE.

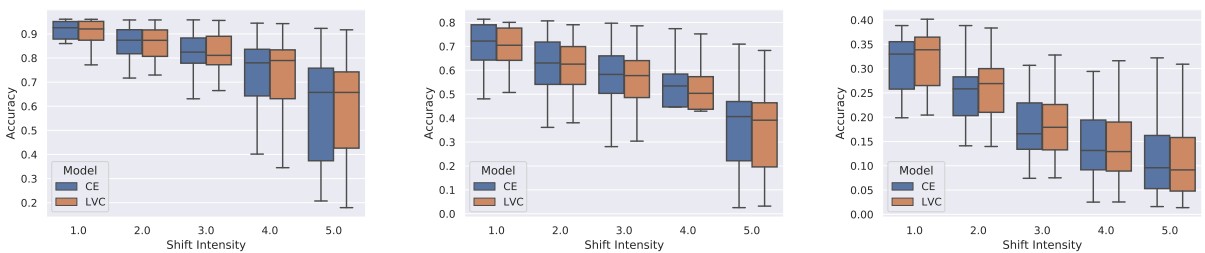

Figure 8: Classification accuracy under distributional shift: *(left)* CIFAR-10-C *(middle) CIFAR-100-C (right)* TINY-IMAGENET-C

