# OpenReview forum: "Variational Classification: A Probabilistic Generalization of the Softmax Classifier"
_TMLR — Accepted by TMLR_

### Review · Reviewer_kNko · 2023-08-15

**Summary Of Contributions:**

This paper proposes a variational classifier to improve the uncertainty estimation and perturbations via a probabilistic approach. The method is similar how variational autoencoder improves the vanilla autoencoder, and numerical results show the advantage of the proposed method over the benchmark algorithms.

**Audience:**

Yes

**Claims And Evidence:**

Yes

**Requested Changes:**

Please consider my comments towards the adversarial robustness, and also revise the writing of this paper. Readers with statistics/probability/GAN background may understand this paper, but it can be potentially hard for other readers to comprehend because the language is not clear.

**Strengths And Weaknesses:**

Strengths:

[1] The proposed idea is intuitively inspiring, and the authors provide sufficient math in the main paper to clearly illustrate their idea.

[2] The numerical results, except for adversarial attack, look promising.

Weaknesses:

One issue is that the performance in adversarial robustness seems to have no great improvement. Adversarial attack used in this paper is a white-box attack, i.e., the attacker knows the model parameters. To overcome adversarial attack, some specific properties, e.g., how far the samples are from the decision boundary, should be addressed. Since the proposed method is tailored for clean generalization but not adversarial robustness, the current presentation in this paper is not sufficient to claim that the proposed method improves adversarial robustness. The out-of-distribution robustness is already a good contribution, so I would suggest the authors to try black-box adversarial attack, or either just drop this adversarial robustness discussion.

Another issue of this paper is the writing. The writing of this paper can be improved. In particular, some senteces are not clear, and some others have no clear connection to others.

[1] Abstract "Our approach offers a novel probabilistic perspective on the highly familiar softmax classification model, to which it relates similarly to how variational and traditional autoencoders relate". The sentence is long and hard to capture its meaning. Please consider simplify it.

[2] Abstract "inherent inconsistency within softmax classification" is not clear. Elaborate a little bit more on the detail.

[3] Throughout the paper, some times the authors use "autoencoder", but some times it becomes "auto-encoder". Please make the word consistent.

[4] Page 2, "it is intractable to learn the parameters of this model by maximising the conditional log likelihood". What does the word "parameters" refer to? Do you mean the neural network weights?

[5] Page 2, "Instead, a lower bound can be maximised, comparable to the evidence lower bound (ELBO), as used to train a variational auto-encoder (VAE)". The sentence is a bit unclear.

My understanding towards this sentence is that, there is a lower bound term used in variational autoencoder, and the authors consider maximizing it. On the other hand, there is another bound called ELBO, and it is used in some other tasks. In such a case, what is the relationship between ELBO, and the one used in common neural network training, as described in the paragraph above this sentence? Is the commonly used method called SCE?

[6] Page 2, "standard softmax cross-entropy (SCE) objective is such a lower bound for specific choices". What does "such" refer to? In addition, for "For this correspondence," what is the logical relationship between the sentences before and after it?

[7] Page 2, "that two versions of the class-conditional latent distributions p(z|y) can be described". Why do we consider two versions? Could you provide one more sentence in this paragraph to quickly mention how to use them? Besides, the writing of whole paragraph needs improvement. The sentence in the middle is short but with four commas, which break the logic of the sentence.

[8] Page 2, "can differ materially". How to understand the word "materially"? Could you use a more accurate word?

[9] Page 2, "In particular, the SCE objective can be optimal if empirical class-conditional latent distributions “collapse” to distinct points rather than fit the anticipated distributions." My understanding is that if something is "optimal", then it will have good properties. But from this sentence, the "optimality" is not sth we expect. Do you mean the the empirical loss achieves the minimal value and overfits the training data?

[10] Page 2, "we take an adversarial approach to implicitly learn the required log probability ratios as an auxiliary task." Could you add one sentence here to briefly introduce what is the adversarial approach? The "adversarial approach" till now appears twice, but there is no explanation about it now. I would suggest to add a short description like "similar to generative adversarial network (GAN)".

[11] Page 2, "The VC framework ..., which the softmax (or more generally output) layer “flips” by Bayes’ rule". Check the grammar of "which".

[12] Page 5, "fill out". Please be more precise on how to understand it.

[13] In the experiment of adversarial robustness, the definition of FGSM is $\epsilon * sign(L)$. Should it be $\epsilon*sign(grad(L))$?

---

> ### Author Response · Authors · 2023-10-17
> **Response from Paper Authors**
>
> Thank you for your review of our work. Please find below detailed responses to the points raised. We attach an updated draft of the paper and have redrafted the abstract and introduction to better clarify the aims and achievements of the paper.
> * **adversarial robustness**:
>     - We would like to clarify that the *core aim of the paper is to improve theoretical understanding of softmax neural network classifiers*. We show that training these classifiers under cross-entropy loss is a special case of training a latent variable model under an ELBO-like objective. That latent variable model ascribes interpretable roles to classifier components and, importantly, shows that the softmax layer implicitly requires its (immediate) inputs to follow a particular distributional family for true class distributions to be output. However, that *anticipated* distribution is neither enforced by the loss function, nor followed empirically. This shows a theoretical inconsistency built into softmax classifiers - which might, at least partially, be to blame for some of their known issues. To test this, we add a loss term to mitigate the discrepancy and so hopefully improve predicted label distributions and properties that rely on them (not only their mode, governing "accuracy"). Fig 1 shows that the distribution of softmax inputs changes empirically as predicted theoretically, and our experiments demonstrate that several known issues with neural network classification are indeed improved - *none of which are "aimed specifically at" by our approach*. We consider this as important context for the results, since if we were targeting a particular issue, e.g. "adversarial robustness", we would need to provide more empirical evidence to show that. Instead, we show that our theoretically derived adjustment to the training objective improves multiple aspects simultaneously, supporting the theory.
>    - We consider it important to show that *without targeting adversarial robustness*, VC is more robust to FGSM perturbation of its own accord. We do not claim to be "robust to all adversarial attacks", and have qualified the claim to avoid such implication, but our results demonstrate that robustness to FGSM perturbations -- an arbitrarily chosen adversarial attack -- improves as a bi-product of the imposed latent structure.
>
> 1,2. **Abstract**: The abstract (and introduction) has been reworded for clarity/readability.
>
> 3. **"auto(-)encoder"**: Thank you, amended.
> 4. **"parameters of this model"**: We refer to *distribution* parameters $\theta, \phi$ of the probabilistic model defined in the text. We have made this more clear.
> 5. **"A lower bound can be maximised, comparable to the evidence lower bound (ELBO), as used to train a VAE"**:
>    - VAEs are trained by maximising the "ELBO", a defined lower bound on the log likelihood for a latent variable model of $p(x)$.
>    - Our training objective is a comparable lower bound on the log likelihood for a latent variable model of $p(y|x)$.
>    - Softmax classifiers are typically trained by minimising Cross-entropy (SCE) loss.
>    - We show that the SCE loss is a special case of our training objective.
>
>    Wording has been revised for clarity.
>
> 6. **"SCE objective is such a lower bound"**:
>    - "such" refers to the family of ELBO-based lower bounds mentioned previously.
>    - Softmax cross-entropy (SCE) loss is a member of that lower bound family, under specific distribution choices, i.e. the lower-bound family generalises SCE.
>    - "For this correspondence," describes the choices needed to show SCE is a special case of the lower bound family.
>
>    The introduction has been revised for readability.
>
> 7.  **"two versions of the class-conditional latent distributions"**: having described the latent variable model for $p(y|x)$, generalising softmax classification, we now show the "inherent inconsistency" in softmax classification: that $p(z|y)$ can be described in two ways, which may differ. The introduction has been revised for readability.
> 8. **"Can differ materially"** - we mean this in the typical sense "can be different to a material, or non-negligible, extent".
> 9. **"In particular, the SCE objective can be optimal if ..."**: "Optimal" refers to when the SCE objective, a function of the data and parameters, is optimised w.r.t. its parameters, i.e. when optimisation stops and the objective reaches an "optimal" value. "Optimality" does not necessarily imply anything we might consider "good"/"bad", although an optimised model *may* have desired properties, e.g. being accurate or calibrated.
> 10. **"we take an adversarial approach":** Introduction has been revised for readability.
> 11. **"The VC framework ..., which the softmax ...."** Grammar confirmed as intended.
> 12. **"fill out"**: We refer, intuitively, to latent variables "fitting a distribution" as opposed to concentrating on a single point (cf MLE/MAP estimates).
> 13. **"definition of FGSM"**: Thank you - amended.

---

### Review · Reviewer_HL7M · 2023-09-05

**Summary Of Contributions:**

The paper proposes a new method to train classification models. The method is motivated by the ideas from variational inference: treating the pre-softmax layer of the classification neural network as an unobserved random variable. A new VC objective is proposed to ensure that the class-conditional distribution of the aforementioned latent variable does not collapse during training. Empirical evaluation demonstrates that the proposed objective results in better adversarial robustness and improves performance in the low-data regime.

**Audience:**

Yes

**Claims And Evidence:**

Yes

**Requested Changes:**

I would like the authors to clarify the following questions:
* What is the motivation to say that variational posterior is equal to $p_{\theta}(z|x)$ (section 3.1.1)? This removes the KL term from the ELBO. As a result, we only have MLE objective. How will the analysis of section 3.3 change if this assumption is not made? Do we still need to align anticipated and empirical distribution in this case?
* $q_{\phi}$ does not seem to be used for variation distribution from section 3.3 onwards, but rather to distinguish anticipated and empirical class conditional distribution. Is my understanding correct?
* Title of the figure 1 refers to $p_{\theta}(z|y)$, is it the same as $q_{\phi}(z|y)$ from equation 9?

Other comments:
* In my opinion, the introduction is too long and contains unnecessary details, which belong to section 3.
* Introduction, paragraph 2: ‘different yet confident predictions elsewhere, which are thus uncertain’. My understanding is that confident and uncertain are two opposite things, how can predictions be both?
* Algorithm 1 does not contain $\beta$.
* Figure 5 requires a more detailed explanation. What does the green line represent? How the locations of the blue points were chosen?

**Strengths And Weaknesses:**

Strengths:
* Interesting analysis regarding the anticipated and empirical class-conditional latent distribution
* Extensive empirical evaluation of the method

Weaknesses:
* Motivation and certain assumptions were not clear to me (see questions below)
* It is not shown how the proposed approach compares to the baseline in terms of the number of parameters and training time.

---

> ### Author Response · Authors · 2023-10-17
> **Response from Paper Authors**
>
> Thank you for your review of our work. Please find below detailed responses to the points raised. We attach an updated draft of the paper and have redrafted the abstract and introduction to better clarify the aims and achievements of the paper.
> * **Comparison to baseline in terms of the number of parameters and training time**: The additional number of parameters is proportional to the number of classes and is small relative to the number of parameters in a typical CNN, e.g. ResNet. In our experiments, wall-clock training time increased for the VC model relative to CE by 10-15%.
> * **Motivation for assuming variational posterior $q(z|\cdot)$ equal to $p(z|x)$**:
>     * Our aim is to gain a clearer mathematical understanding of softmax classification by deriving a latent variable model that generalises a softmax classifier, and a training objective that generalises softmax cross-entropy (SCE). Such a generalisation attributes semantic meaning to the components of a softmax classifier, and any conclusions drawn from it immediately apply to softmax classification.
>     * §3 derives a training objective to learn parameters of the latent prediction model (Eq 6, *now 5*).
>     * Due to intractability of the likelihood, we use an ELBO-like objective in line 1, Eq 7 (*now 6*). (This is equivalent to the standard ELBO for $\log p(y)$, conditioned throughout on $x$). The variational distribution $q_\phi(z|\cdot)$ can be freely chosen, including anything it is conditioned on, so $q_\phi(z|\cdot)=p_\theta(z|x)$ is a valid choice (from the KL term, we know that as the overall objective is maximised $q_\phi(z|\cdot)$ approximates $p_\theta(z|x,y)$ as well as possible). Importantly for our analysis, this choice also gives an objective that generalises the SCE loss, so that we have a generalisation of both a softmax classifier and its training objective, as aimed for (i.e. assumptions can be plugged in to give the SCE loss).
>     * The result is indeed an "MLE objective", which necessarily implies that SCE is also.
>     * This choice is to generalise SCE loss. Another choice may not achieve that and enable the subsequent analysis and conclusions.
>     * By making this choice, showing that $ELBO_{VC}$ generalises SCE, and seeing that the empirical and anticipated distributions may differ, we are able to conclude that this may be an issue that affects softmax classification.
> * **$q$ does not seem to be used for variation distribution**: $q_\phi(z|x)$, and $q_\phi(z|y)$ are latent distributions parameterised by the encoder network; $p_\theta(z|y)$ is an analytically specified latent distribution encoded in the (generalised) softmax layer. (See also footnote 4, p.4)
> * **Title of the figure 1 refers to $p(z|y)$**: Thank you, updated to "$q_\phi(z|y)$"
> * **"Intro too long", "different yet confident predictions elsewhere, which are thus uncertain"** - we refer to samples for which a model appears confident when it *ought* to be uncertain. Introduction has been made shorter and revised for readability.
> * **Algorithm 1 does not contain $\beta$**: Thank you, the algorithm has been updated.
> * **Figure 5 requires a more detailed explanation**: Figure 5 shows the improvement of VC over CE for 10 datasets from the Med-MNIST collection, which vary in the number of data samples. The green dashed line shows a smoothed trend to illustrate how the benefit of VC increases for smaller datasets. The caption has been updated.

---

### Review · Reviewer_zQao · 2023-09-26

**Summary Of Contributions:**

In this paper, the authors propose to enhance softmax classification done via deterministic neural networks with a variational approach to the latent space. By introducing a probabilistic model of the latent space, the authors propose to regularize the softmax classification model in order to align the empirical and expected latent distribution, the resulting objective is termed “Variational Classification.” This approach is aimed specifically at improving model calibration, reducing data complexity, increasing distributional robustness, and increasing adversarial robustness when compared with deterministic neural networks. Though I have evaluated the authors high-level formalization, I was unable to perform a thorough assessment of proofs in the Appendix, and may have missed details therein. Thus, the critical components of my review focuses primarily on two aspects of the paper (1) the related work and (2) the empirical evaluation.

**Audience:**

Yes

**Broader Impact Concerns:**

I have no specific broader impact concerns for this work.

**Claims And Evidence:**

No

**Requested Changes:**

* [Critical] It seems straight-forward to me that the hypothetical improvements that this method seeks to achieve are also hypothetical improvements of other probabilistic approaches to machine learning, namely Bayesian neural networks (BNNs). It has been shown in various works that Bayesian neural networks are more resilient to natural data distribution shifts as well as to adversarial perturbations. Moreover their principled uncertainty serves as a key component of active learning approaches which is a tried and true method for reducing sample complexity in supervised learning. Yet, the authors do not discuss any of these works or even that this line of literature exists. I find this a bit peculiar, and would like for the authors to justify this choice. It is clear the method they propose scales much better than faithful approximate posterior inference, however presenting no such discussion feels like a significant omision.

* [Non-Critical] Perhaps, asking for comparison with other probabilistic neural networks is misguided; however, I would also like to see a simple comparison with NoisyAdam or MC Dropout to understand how this method performs in comparison to other probabilistic approaches. I would like to see the empirical comparison across all of the hypotheses listed by the authors given that these are areas in which BNNs purport to give some advantages

* [Non-critical] In the authors evaluation, I find it a bit strange that the GM method performs worse in terms of calibration across almost all datasets. Could the authors discuss why this might be the case? Should their method not improve the model’s calibration?

*[Critical] The experiments on adversarial robustness are far too scant to be useful to any reader. It is also too rudimentary to serve as evidence for the claim of (H3). For example, the authors have provided an epsilon, but no indication of which norm/metric this epsilon is with respect to (I assume l2 given the magnitude of the values) but then again, l_\infty is a much more common choice. Moreover, the authors do not evaluate with any adaptive attack despite this being a clear standard in the literature. I understand that perhaps FGSM is an initial indicator of improved adversarial robustness but it is hardly practically significant. The authors need to either considerably increase the rigor, detail, and experimental rational of this section of entirely remove it and remove any claim of (H3) because as it stands, it is not supported by theory or by experimental evidence.


Minor weaknesses

* Please label each of the subplots of figure 3 so that it is easily digestible

**Strengths And Weaknesses:**

* The paper studies studies the critical problem of improving the reliability of deep neural networks.

* The communication of the paper, its central claims, and methods is clear and concise with high-quality expositional figures.

* Though lacking in some ways, the experimental design is sufficient to corroborate the potential of the method. I find the experiments on generalization under distribution shift particularly compelling.

---

> ### Author Response · Authors · 2023-10-17
> **Response from Paper Authors**
>
> Thank you for your review of our work. Please find below detailed responses to the points raised. We attach an updated draft of the paper and have redrafted the abstract and introduction to better clarify the aims and achievements of the paper.
> * **"approach is aimed specifically at improving model calibration, data complexity, distributional robustness, and adversarial robustness"**:
>    - We would like to clarify that the core aim of the paper is *to improve theoretical understanding of softmax neural network classifiers*. We show that training these classifiers under cross-entropy loss is a special case of training a latent variable model under an ELBO-like objective. The latent variable model ascribes interpretable roles to classifier components and, importantly, shows that the softmax layer implicitly requires its (immediate) inputs to follow a particular distributional family for true class distributions to be output. However, this *anticipated* input distribution is neither enforced by the loss function, nor followed empirically. This is of interest as it shows a theoretical inconsistency built into softmax classifiers - the implication being that it might be, at least partially, to blame for some of their known issues. We add a loss term to mitigate the discrepancy, which is expected to improve predicted label distributions (not only their mode, governing "accuracy"). We demonstrate (Fig 1) that the distribution of softmax inputs changes empirically as expected theoretically, and experimentally demonstrate that several known issues with neural network classification are indeed improved - *none of which are "aimed specifically at" by our approach*. We consider this as important context in which to consider the results, since if we were targeting a particular issue, e.g. adversarial robustness, we would need to provide more empirical evidence to show that. Instead, we show that our theoretically derived adjustment to the training objective improves multiple aspects simultaneously, supporting the theory.
> * **Adversarial robustness**: We consider it important to show that *without targeting adversarial robustness*, VC is more robust to FGSM perturbation of its own accord. We do not claim to be "robust to all adversarial attacks", and have qualified the claim to avoid this implication. Our results demonstrate that robustness to FGSM perturbations - an arbitrarily chosen adversarial attack - improves as a bi-product of the imposed latent structure. For comparability, FGSM experiments (and the range of $\epsilon$ values) follow [3 (Table 6)], where $\epsilon$ is a scalar that defines the $\ell_\infty$ norm of the perturbation vector.
> * **Bayesian Neural Networks**: As above, our core aim is to better understand classifiers from a *latent variable* perspective, whereas Bayesian Neural Networks are typically Bayesian with respect to *parameters*, an orthogonal consideration. A Bayesian approach can be taken to both latent variables and parameters, e.g. VBEM [1]. Applying a distribution to encoder weights, as in BNNs or MC dropout (Gal et al., 2016), effectively increases the variance of $q_\phi(z|y)$ in our latent model, and could be interesting to explore in future research.
> We include accuracy and ECE results for MC dropout on the CIFAR datasets to contextualise the impact of VC. It can be seen that while MC dropout does improve calibration, critically (and further to its increased computational cost) it *loses accuracy* relative to CE, as observed previously [2].
> * **GM model calibration results**: The GM model is equivalent to softmax cross entropy with a prior over the latent variables. Maximising this objective results in a MAP *point estimate* of the latent variables (albeit a different point to the MLE case), and all latents of a class "collapse" to a point, which loses information and is not expected to improve calibration. By comparison (to both CE and GM), the VC model includes an entropy term, which causes the $q_\phi(z|y)$ distributions to "puff out" (w.r.t. a point) and "fit" $p_\theta(z|y)$ (see Fig 1), reducing over-confidence of predictions $p(y|x)$ and improving calibration.
>
> [1] Murphy, K.P., 2012. Machine learning: a probabilistic perspective. MIT press
>
> [2] Ovadia et al., 2019. Can you trust your model's uncertainty? evaluating predictive uncertainty under dataset shift. NeurIPS
>
> [3] Wan et al., 2018. Rethinking feature distribution for loss functions in image classification. CVPR

---

### Decision · Action_Editor_A9jP · 2023-12-01

**Recommendation:** Accept as is

**Comment:**

This is a strong paper with a new and fresh framework for classification via latent variable models. The basic idea (generalizing standard softmax classifiers to a latent variable model that can be trained with an ELBO-like approach) is simple but interesting, and leads to some nice properties, like the calibration finding the authors show.

This is solid research that is worth accepting.

All reviewers agree that the work is genuinely interesting. The main questions raised (resolved by the authors during the discussion phase) relate to the exact positioning of the work, and the reviewers were convinced.

Beyond this, there were requests to improve the writing, which the authors have done.

**Audience:**

Yes, frameworks that offer a new take on standard components of the machine learning pipeline are always of interest.

**Claims And Evidence:**

Yes, the paper does a good job marshalling both theoretical arguments (i.e., standard softmax is a special case of their variational framework) and empirical validation.